# Receptor-binding loops in alphacoronavirus adaptation and evolution

Alan H.M. Wong[1], Aidan C.A. Tomlinson[1], Dongxia Zhou[2], Malathy Satkunarajah[2], Kevin Chen[1], Chetna Sharon[2], Marc Desforges[3], Pierre J. Talbot[3] & James M. Rini[1,2]

RNA viruses are characterized by a high mutation rate, a buffer against environmental change. Nevertheless, the means by which random mutation improves viral fitness is not well characterized. Here we report the X-ray crystal structure of the receptor-binding domain (RBD) of the human coronavirus, HCoV-229E, in complex with the ectodomain of its receptor, aminopeptidase N (APN). Three extended loops are solely responsible for receptor binding and the evolution of HCoV-229E and its close relatives is accompanied by changing loop–receptor interactions. Phylogenetic analysis shows that the natural HCoV-229E receptor-binding loop variation observed defines six RBD classes whose viruses have successively replaced each other in the human population over the past 50 years. These RBD classes differ in their affinity for APN and their ability to bind an HCoV-229E neutralizing antibody. Together, our results provide a model for alphacoronavirus adaptation and evolution based on the use of extended loops for receptor binding.

[1] Department of Biochemistry, University of Toronto, 1 King's College Circle, Toronto, Ontario, Canada M5S 1A8. [2] Department of Molecular Genetics, University of Toronto, 1 King's College Circle, Toronto, Ontario, Canada M5S 1A8. [3] Laboratory of Neuroimmunovirology, INRS-Institut Armand-Frappier, Institut National de la Recherche Scientifique, Université du Québec, 531 Boulevard des Prairies, Laval, Québec, Canada H7V 1B7. Correspondence and requests for materials should be addressed to J.M.R. (email: james.rini@utoronto.ca)

Coronaviruses are enveloped, positive-stranded RNA viruses that cause a number of respiratory, gastrointestinal, and neurological diseases in birds and mammals[1, 2]. The coronaviruses all possess a common ancestor and four different genera (alpha, beta, gamma, and delta) that collectively use at least four different glycoproteins and acetylated sialic acids as host receptors or attachment factors have evolved[3–5]. Four coronaviruses, HCoV-229E, HCoV-NL63, HCoV-OC43, and HCoV-HKU1 circulate in the human population and collectively they are responsible for a significant percentage of the common cold as well as more severe respiratory disease in vulnerable populations[6, 7]. HCoV-229E and HCoV-NL63 are both alphacoronaviruses and although closely related, they have evolved to use two different receptors, aminopeptidase N (APN) and angiotensin converting enzyme 2 (ACE2), respectively[8, 9]. The more distantly related betacoronaviruses, HCoV-OC43 and HCoV-HKU1, are less well characterized and although HCoV-OC43 uses 9-O-acetylsialic acid as its receptor[10], the receptor for HCoV-HKU1 has not yet been determined[11–13]. Recent zoonotic transmission of betacoronaviruses from bats is responsible for SARS and MERS, and in these cases infection is associated with much more serious disease and high rates of mortality[14–16]. Like HCoV-NL63, SARS-CoV uses ACE2[17] as its receptor and the observation that MERS-CoV uses dipeptidyl peptidase 4[18] highlights the fact that coronaviruses with new receptor specificities continue to arise.

The coronavirus spike protein (S-protein) is a trimeric single-pass membrane protein that mediates receptor binding and fusion of the viral and host cell membranes[19]. It is a type-1 viral fusion protein possessing two regions, the S1 region that contains the receptor-binding domain (RBD) and the S2 region that contains the fusion peptide and heptad repeats involved in membrane fusion[20–25]. The coronavirus S-protein is also a major target of neutralizing antibodies and one outcome of host-induced neutralizing antibodies is the selection of viral variants capable of evading them, a process known to drive variation[26–28]. As shown by both in vivo and in vitro studies, changes in host, host cell type, cross-species transmission, receptor expression levels, serial passage, and tissue culture conditions can also drive viral variation[29–33].

RNA viruses are characterized by a high mutation rate, a property serving as a buffer against environmental change[34]. A host-elicited immune response, the introduction of antiviral drugs, and the transmission to a new species provide important examples of environmental change[35]. Nevertheless, the means by which random mutations lead to viral variants with increased fitness and enhanced survival in the new environment are not well characterized. Given their wide host range, diverse receptor usage and ongoing zoonotic transmission to humans, the coronaviruses provide an important system for studying RNA virus adaptation and evolution. The alphacoronavirus, HCoV-229E, is particularly valuable as it circulates in the human population and a sequence database of natural variants isolated over the past fifty years is available. Moreover, changes in sequence and serology have suggested that HCoV-229E is changing over time in the human population[36–38].

Reported here is the X-ray structure of the HCoV-229E RBD in complex with human APN (hAPN). The structure shows that receptor binding is mediated solely by three extended loops, a feature shared by HCoV-NL63 and the closely related porcine respiratory coronavirus, PRCoV. It also shows that the HCoV-229E RBD binds at a site on hAPN that differs from the site where the PRCoV RBD binds on porcine APN (pAPN), evidence of an ability of the RBD to acquire novel receptor interactions. Remarkably, we find that the natural HCoV-229E sequence variation observed over the past fifty years is highly skewed to the receptor-binding loops. Moreover, we find that the loop variation defines six RBD classes (Classes I–VI) whose viruses have successively replaced each other in the human population. These RBD classes differ in their affinity for hAPN and their ability to be bound by a neutralizing antibody elicited by the HCoV-229E reference strain (Class I). Taken together, our results provide a model for alphacoronavirus adaptation and evolution stemming from the use of extended loops for receptor binding.

## Results

**Characterization of the HCoV-229E RBD interaction with hAPN.** To define the limits of the HCoV-229E RBD, we expressed a series of soluble S-protein fragments and measured their affinity to a soluble fragment (residues 66–967)[39] of hAPN, the HCoV-229E receptor. The smallest S-protein fragment made (residues 293–435) bound hAPN with an affinity ($K_d$ of $0.43 \pm 0.1\,\mu M$) similar to that of the entire S1 region (residues 17–560) (Table 1, Supplementary Fig. 1A, B) and this fragment was used in the structure determination. To confirm the importance of the

**Table 1 Analysis of the hAPN ectodomain (residues 66–967, WT and mutants) interaction with fragments of the HCoV-229E S-protein (WT and mutants) using surface plasmon resonance**

| HCoV-229E | $k_{on}$ (×10⁵ M⁻¹ s⁻¹) | $k_{off}$ (s⁻¹) | $K_d$ (μM) |
|---|---|---|---|
| 17-560 (S1) WT | $0.39 \pm 0.03$ | $0.06 \pm 0.02$ | $1.63 \pm 0.17$ |
| 293-435 (RBD) WT | $3.6 \pm 0.53$ | $0.16 \pm 0.02$ | $0.43 \pm 0.06$ |
| 293-435 (RBD) F318A | $1.4 \pm 0.15$ | $0.84 \pm 0.06$ | $5.8 \pm 0.05$ |
| 293-435 (RBD) N319A | — | — | n.b. at 25 μM |
| 293-435 (RBD) W404A | — | — | n.b. at 2.2 μM |
| 293-435 (RBD) C317S/C320S (double mutant) | — | — | n.b. at 15 μM |

| hAPN | $k_{on}$ (×10⁵ M⁻¹ s⁻¹) | $k_{off}$ (s⁻¹) | $K_d$ (μM) |
|---|---|---|---|
| WT hAPN | $3.6 \pm 0.53$ | $0.16 \pm 0.02$ | $0.43 \pm 0.06$ |
| hAPN D288A | $1.4 \pm 0.32$ | $0.67 \pm 0.20$ | $4.6 \pm 0.35$ |
| hAPN Y289A | $1.3 \pm 0.27$ | $1.0\ \pm 0.1$ | $7.8 \pm 0.71$ |
| hAPN V290G | $6.0 \pm 1.27$ | $0.74 \pm 0.01$ | $12.8 \pm 2.8$ |
| hAPN I309A | $1.4 \pm 0.4$ | $1.45 \pm 0.22$ | $10.7 \pm 1.8$ |
| hAPN L318A | $2.8 \pm 0.3$ | $1.43 \pm 0.30$ | $5.2 \pm 0.92$ |
| hAPN E291N/K292E/Q293T (triple mutant) | — | — | n.b. at 8 μM |

n.b. no binding
Values after ± correspond to the residual standard deviation reported by Scrubber 2. Two experiments were performed

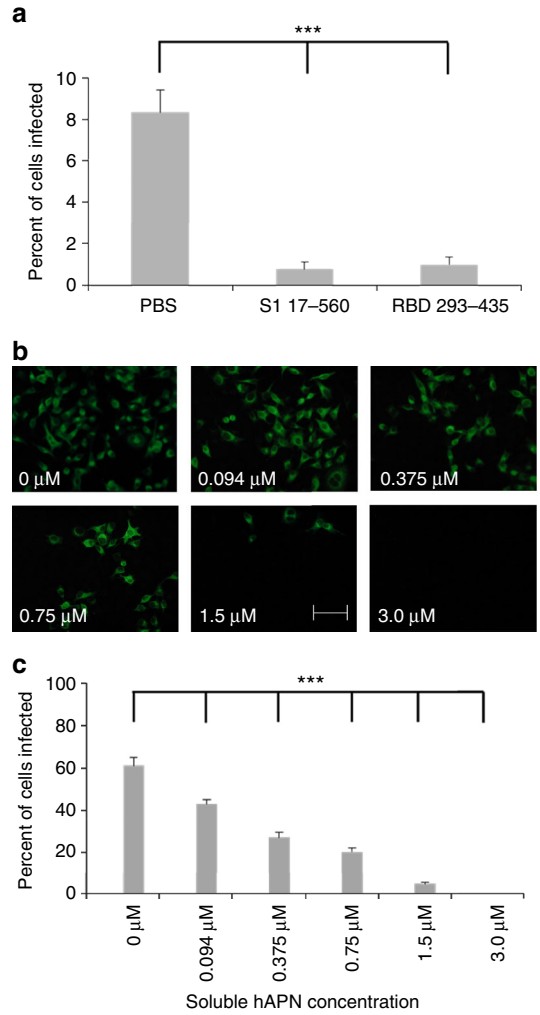

| Table 2 X-ray crystallographic data collection and refinement statistics | |
| --- | --- |
| | **HCoV-229E RBD–hAPN** |
| Data collection | |
| Space group | P3$_1$21 |
| *Cell dimensions* | |
| *a, b, c* (Å) | 153.8, 153.8, 322.1 |
| *α, β, γ* (°) | 90, 90, 120 |
| Wavelength(Å) | 0.9795 |
| Resolution (Å) | 50–3.5 (3.6–3.5) |
| No. of total reflections | 229,646 (22,754) |
| No. of unique reflections | 55,987 (5490) |
| CC$_{1/2}$ | 99.1 (68.1) |
| CC$^*$ | 99.8 (90) |
| $R_{sym}$ | 0.16 (0.70) |
| $R_{pim}$ | 0.08 (0.33) |
| $I/\sigma I$ | 10.9 (2.7) |
| Completeness (%) | 99.6 (99.8) |
| Redundancy | 4.1 (4.2) |
| Refinement | |
| Resolution (Å) | 50–3.5 |
| No. of reflections | 55,969 |
| $R_{work}/R_{free}$ | 0.24 (0.31) /0.27 (0.32) |
| *No. of atoms* | |
| Protein | 23306 |
| N-glycans | 353 |
| Water | 0 |
| B-factors (Å$^2$) | |
| Protein | 102 |
| N-glycans | 110 |
| Wilson B-value (Å$^2$) | 95 |
| R.m.s. deviations | |
| Bond lengths (Å) | 0.004 |
| Bond angles (°) | 0.71 |
| Ramachandran stats. (%) | |
| Favored | 97 |
| Outlier | 0 |

Values in parenthesis are for the highest resolution shell

**Fig. 1** Characterization of soluble fragments of the HCoV-229E S-protein and hAPN. **a** HCoV-229E infection of L-132 cells in the presence of: PBS, the HCoV-229E S1 domain (residues 17–560 at 10 μM), and the HCoV-229E RBD (residues 293–435 at 30 μM). Statistics were obtained from three independent experiments. Statistical significance (ANOVA): ***$p < 0.001$; error bars correspond to the standard deviation. **b** Representative images of HCoV-229E infection of L-132 cells in the presence of the hAPN ectodomain at various concentrations. Green fluorescence measures the expression of the viral S-protein. Magnification (100×) and scale bar = 20 μm. **c** Quantitation of the hAPN inhibition experiment. Statistics were obtained from three independent experiments. Statistical significance (ANOVA): ***$p < 0.001$

HCoV-229E RBD–hAPN interaction for viral infection, we showed that both the RBD and the hAPN ectodomain inhibited viral infection in a cell-based assay (Fig. 1a, b, c).

Crystals of the HCoV-229E RBD–hAPN complex were obtained by co-crystallization of the complex after size exclusion chromatography. The crystallographic data collection and refinement statistics are shown in Table 2. The asymmetric unit contains one hAPN dimer (and associated RBDs) and one hAPN monomer (and associated RBD) that is related to its dimeric mate by a crystallographic two-fold rotation axis. Both dimers (non-crystallographic and crystallographic) are found in the closed conformation and are essentially identical to that which we previously reported[39] for hAPN in its *apo* form (RMSD over all Cα atoms of 0.34 Å). Each APN monomer is bound to one RBD as shown in Fig. 2a. The HCoV-229E RBD–hAPN interaction buries 510 Å$^2$ of surface area on the RBD and 490 Å$^2$ on hAPN.

The HCoV-229E RBD is an elongated six-stranded β-structural domain with three extended loops (loop 1: residues 308–325, loop 2: residues 352–359, loop 3: residues 404–408) at one end that exclusively mediate the interaction with hAPN (Fig. 2b). Loop 1 is the longest and it contributes ~70% of the RBD surface buried on complex formation (Figs. 2c and 3g). Within loop 1, residues Cys$^{317}$ and Cys$^{320}$ form a disulfide bond that makes a stacking interaction with the side chains of hAPN residues Tyr$^{289}$ and Glu$^{291}$ (Fig. 2c). The C317S/C320S RBD double mutant showed no binding to hAPN at concentrations up to 15 μM (Table 1, Supplementary Fig. 1D, and Supplementary Table 1), evidence of the importance of the stacking interaction and a likely role for the disulfide bond in defining the conformation of loop 1. Notably, loop 1 contains three tandemly repeated glycine residues (residues 313–315) whose NH groups donate hydrogen bonds to the side chain of Asp$^{288}$ and the carbonyl oxygen of Phe$^{287}$ of hAPN (Fig. 2c); mutation of hAPN residue Asp$^{288}$ to alanine leads to a ~10-fold reduction in affinity (Table 1, Supplementary Fig. 2A, and Supplementary Table 1). Apolar interactions between RBD residues Cys$^{317}$ and Phe$^{318}$ and hAPN residues Tyr$^{289}$, Val$^{290}$, Ile$^{309}$, Ala$^{310}$, and Leu$^{318}$ are also observed (Fig. 2c); mutation of RBD residue Phe$^{318}$ leads to a 13-fold reduction in affinity while mutation of hAPN residues Tyr$^{289}$, Val$^{290}$, Ile$^{309}$, and Leu$^{318}$ lead to a 10- to 30-fold reduction in affinity (Table 1, Supplementary Fig. 1C, Supplementary Fig. 2B–E, and Supplementary Table 1). Centered in the contact

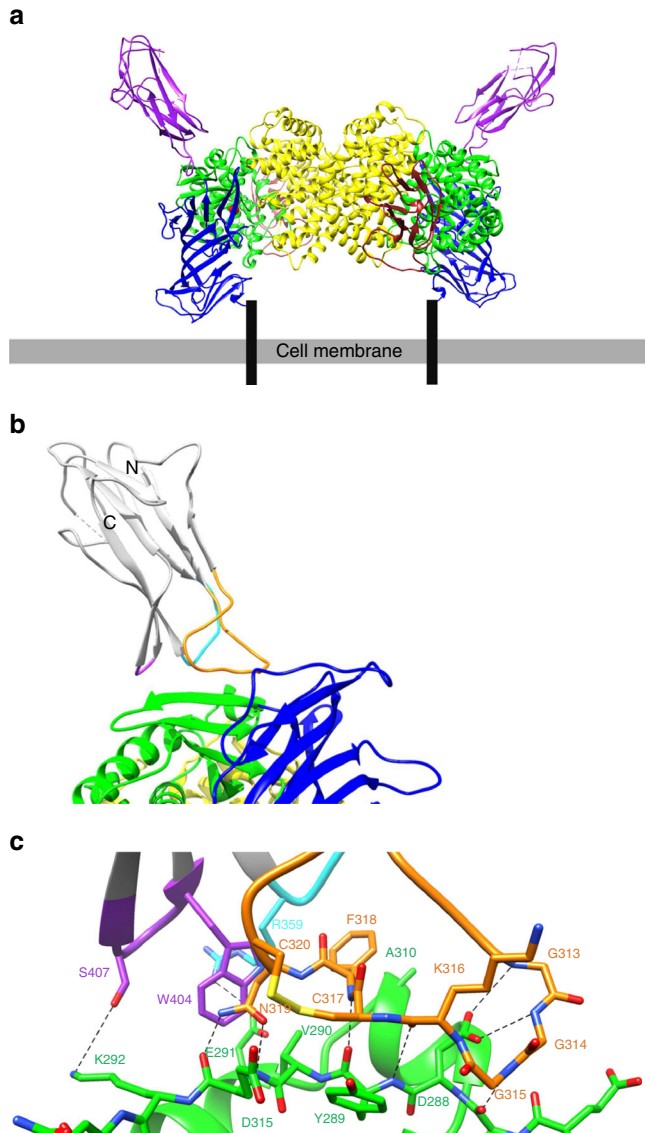

**Fig. 2** HCoV-229E RBD in complex with the ectodomain of hAPN. **a** The complex between dimeric hAPN (domain I: blue, domain II: green, domain III: brown, and domain IV: yellow) and the HCoV-229E RBD (purple) is depicted in its likely orientation relative to the plasma membrane. The hAPN peptide and zinc ion (red spheres) binding sites are located inside a cavity distant from the virus binding site. Black bars represent the hAPN N-terminal transmembrane region. **b** Ribbon representation of the HCoV-229E RBD (gray) in complex with hAPN (same coloring as in **a**). The three receptor-binding loops are colored, orange (loop 1), cyan (loop 2), and purple (loop 3). N and C label the N- and C-termini of the RBD. **c** Atomic details of the interaction at the binding interface. Hydrogen bonds and salt bridges are indicated by dashed lines. Red and blue correspond to oxygen and nitrogen atoms, respectively. Loop and hAPN coloring as in **b**

interactions made by loop 3 residues Trp[404] and Ser[407] with hAPN residues Asp[315] and Lys[292] (Fig. 2c); the importance of Trp[404] of loop 3 is evidenced by the fact that mutating it also ablates binding (Table 1, Supplementary Fig. 1F, and Supplementary Table 1).

**HCoV-229E and PRCoV bind at different sites on APN.** As with HCoV-229E, the porcine respiratory alphacoronavirus, PRCoV, also uses APN as its receptor[40]. As our complex shows, HCoV-229E binds at a site on hAPN (H-site) that differs from the site on pAPN (P-site) used by PRCoV (Fig. 3a, b). Glu[291] in hAPN, a residue in the hAPN–RBD interface, is an N-glycosylated asparagine (Asn[286]) in pAPN and attempts to dock the HCoV-229E RBD at the H-site on pAPN leads to a steric clash with the N-glycan (Supplementary Fig. 3A). Consistent with this observation, the HCoV-229E RBD cannot bind to a mutant form of hAPN (E291N/K292E/Q293T) that possesses an N-glycan at position 291, as we have shown (Table 1, Supplementary Fig. 4A–C). Attempts to dock the PRCoV RBD at the P-site on hAPN also leads to a steric clash, in this case with hAPN residue Arg[741] (Supplementary Fig. 3B). Notably, porcine transmissible gastroenteritis virus (TGEV) can bind hAPN, and HCoV-229E can bind mouse APN, once the Arg side chain (on hAPN) and the N-glycan (on mouse APN) on the respective APNs have been mutated[41]. Across species, the sequence identity at the H- and P-sites is only ~60% (Fig. 3c and Supplementary Fig. 3C) and the receptor-binding loops of these viruses must be accommodating the remaining APN structural differences on receptors from species that they do not infect. Together these results provide evidence that the extended receptor-binding loops of these alphacoronaviruses possess conformational plasticity.

The observation that HCoV-229E and PRCoV bind to different sites on APN has important consequences. Among species, APN is found in open/intermediate and closed conformations and conversion between them is thought to be important for the catalysis of its substrates[39, 42]. The HCoV-229E RBD binds to hAPN in its closed conformation and structural comparison shows that the H-site does not differ between the open and closed conformations. This is to be contrasted with the P-site of pAPN that differs in the open and closed conformations. Indeed, the PRCoV RBD has recently been shown to bind to pAPN in the open conformation as a result of P-site interactions made possible in the open form[42]. These differences in binding and receptor conformation are reflected in the fact that enzyme inhibitors that promote the closed conformation of APN block TGEV infection[42], but not HCoV-229E infection[8], and the fact that the PRCoV S-protein[42], but not HCoV-229E[43], inhibits APN catalytic activity.

**The receptor-binding loops of HCoV-229E vary extensively.** Sequence data from viruses isolated over the past 50 years provides a wealth of data on the natural variation shown by HCoV-229E (Supplementary Fig. 5). With reference to the HCoV-229E RBD–hAPN complex reported here, we now show that 73% of the amino acids in the receptor-binding loops and supporting residues vary among the sequences analyzed (52 sequences in total), while only 11% of the RBD surface residues outside of the receptor-binding loops show variation (Fig. 4a, b). Moreover, for the eight variants where full genome sequences were reported, the receptor-binding loops represent the location at which the greatest variation in the entire genome is observed (Fig. 4c). Analysis of the HCoV-229E RBD–hAPN interface further shows that of the 16 RBD surface residues that are fully or partially buried on complex formation, 10 of them vary in at least one of the 52 sequences analyzed and a pairwise comparison of the

area between the RBD and hAPN is a hydrogen bond between the side chain of RBD residue Asn[319] and the carbonyl oxygen of hAPN residue Glu[291] (Fig. 2c); mutation of RBD residue Asn[319] to alanine also ablates binding at the highest concentrations achievable (Table 1, Supplementary Fig. 1E, and Supplementary Table 1). The remaining loop 1 residues serve to satisfy most of the hydrogen bond donor/acceptor pairs of the edge β-strand on subdomain 2 of the hAPN molecule. Most prominent of the remaining RBD–hAPN interactions is the salt bridge between loop 2 residue Arg[359] and hAPN residue Asp[315] and the

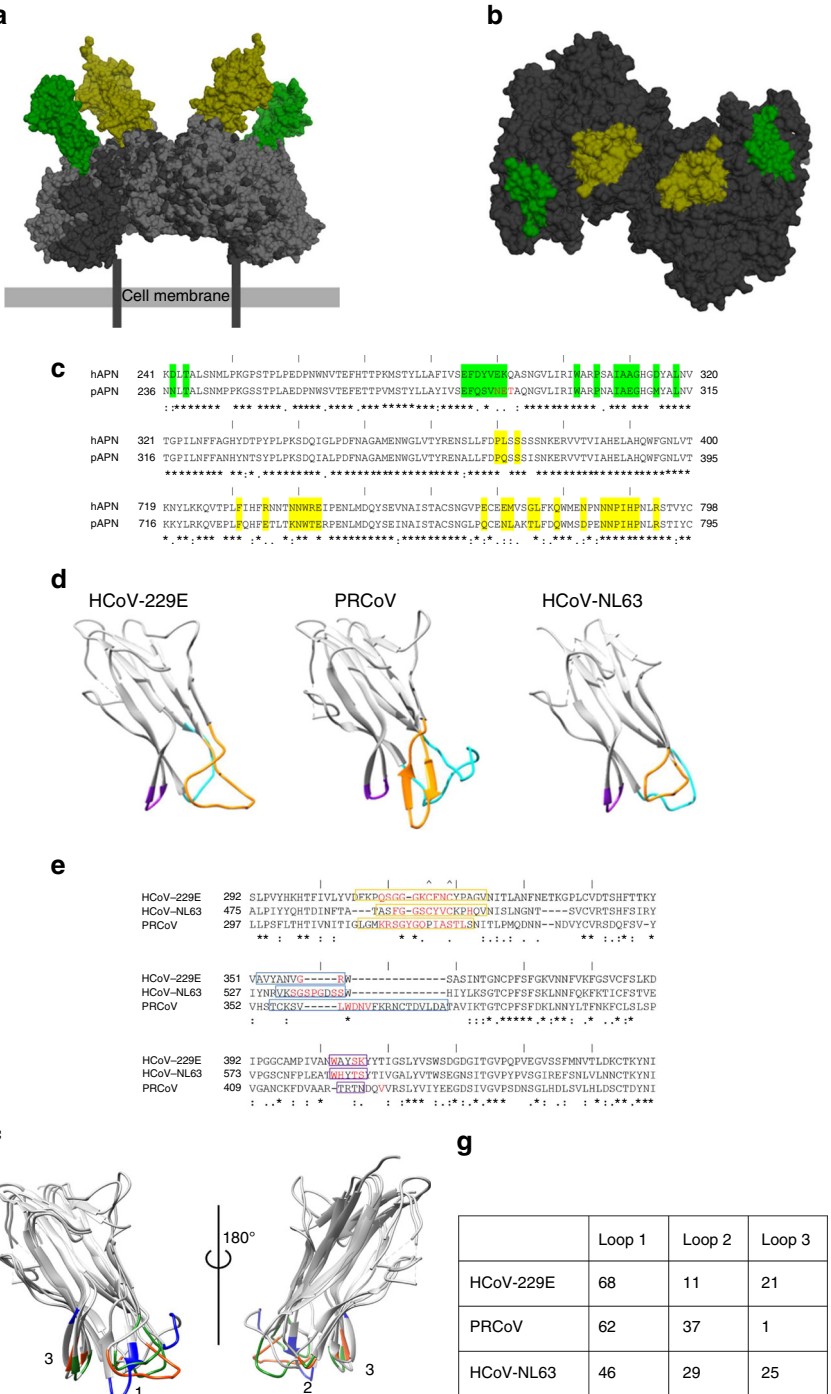

**Fig. 3** Alphacoronavirus receptor-binding domains. **a** Surface representation of an APN-based overlay of the HCoV-229E RBD–hAPN and PRCoV RBD–pAPN complexes. Human APN (dark gray), porcine APN (light gray), HCoV-229E RBD (green), and PRCoV RBD (yellow). APNs are aligned on domain IV. **b** Top view of the APN surface buried by HCoV-229E RBD binding (H-site, green) and PRCoV RBD binding (P-site; yellow) mapped onto hAPN. **c** Sequence alignment of human and porcine APN. Residues in the H-site are highlighted in green and residues in the P-site are highlighted in yellow. The "I" symbol demarcates every 10 residues in the alignment. The *N*-glycosylation sequon (Asn residue 286) in porcine APN is shown in red (Glu residue 291 in human). **d** Ribbon representation of the HCoV-229E RBD (receptor: hAPN), the PRCoV RBD (receptor: pAPN), and the HCoV-NL63 RBD (receptor: hACE2). Loops 1, 2, and 3 are colored in orange, cyan, and purple, respectively. **e** Sequence alignment of the HCoV-229E, PRCoV, and HCoV-NL63 RBDs. Residues in loops 1, 2, and 3 are enclosed by orange, cyan, and purple boxes, respectively. The cysteine residues involved in the loop 1 disulfide bond are indicated by "^". The "I" symbol demarcates every 10 residues in the alignment. Residues directly interacting with the receptor are colored red. **f** Structural alignment of the HCoV-229E, HCoV-NL63, and PRCoV RBDs with receptor interacting residues colored orange, green, and blue, respectively. Numbers indicate the loop numbers. The structures are shown in two views rotated by 180° relative to each other. **g** The percentage contribution made by each loop to the total surface area buried on the RBD in the receptor complexes

sequences suggests that many of these positions can vary simultaneously (Supplementary Fig. 5). Finally, we show that the six invariant interface residues on the RBD (Gly$^{313}$, Gly$^{315}$, Cys$^{317}$, Cys$^{320}$, Asn$^{319}$, and Arg$^{359}$) constitute only 45% of the viral surface area buried, the very region expected to be the most highly conserved from a receptor-binding standpoint. The

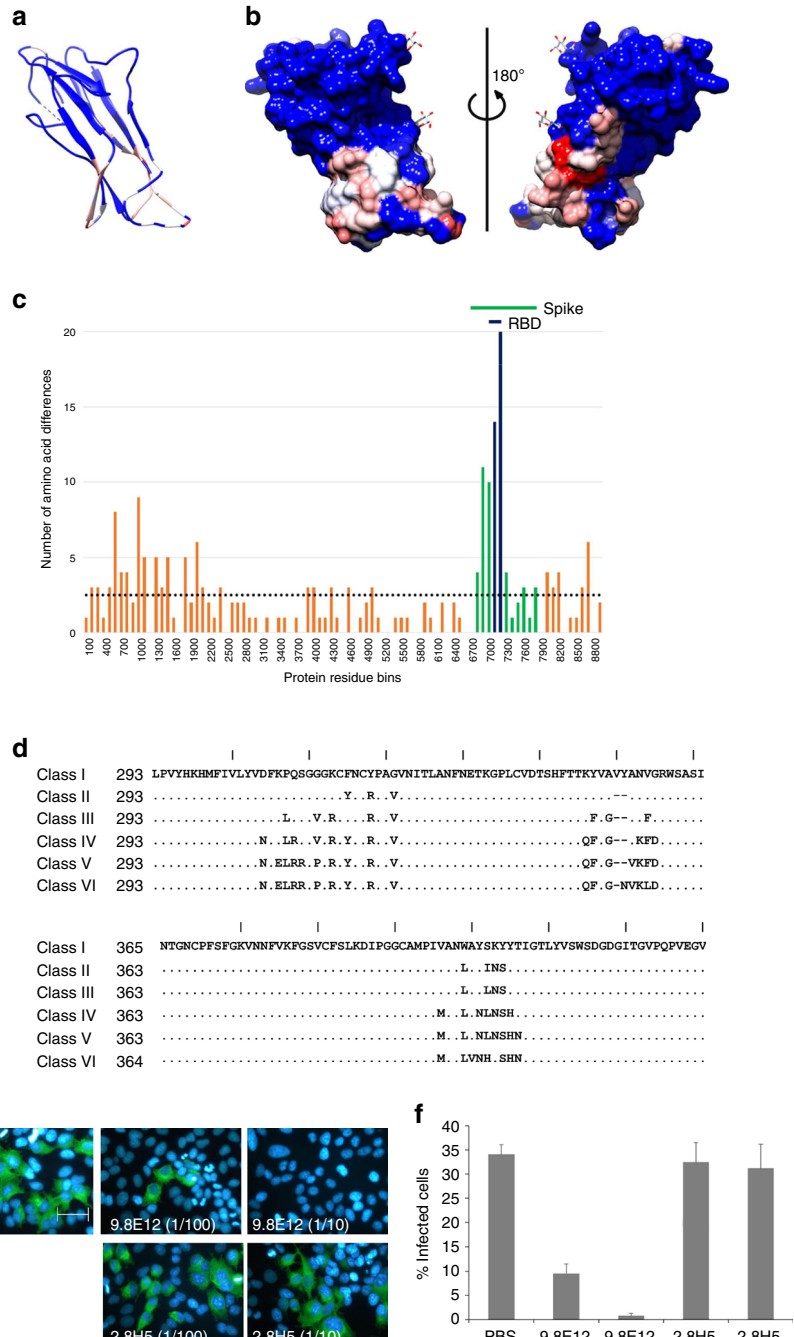

**Fig. 4** Naturally occurring HCoV-229E sequence variation. **a** Color-coded amino-acid sequence conservation index (Chimera) mapped onto a ribbon representation of the HCoV-229E RBD. Blue represents a high percentage sequence identity and red represents a low percentage sequence identity among the 52 viral isolates analyzed. **b** Surface representation in the same orientation as in (**a**, left), and rotated 180° (right). The Asn-GlcNAc moiety of the *N*-glycans are shown in stick representation. Color coding as in **a**. **c** Amino-acid sequence variation shown by the eight viral isolates whose entire genome sequences have been reported. The entire protein coding region of the viral genome was treated as a continuous amino acid string (8850 residues in total). Amino acid differences among the eight sequences were analyzed in 100 residue bins and for each bin the sum was plotted. Green-colored bins correspond to residues in the S-protein and purple-colored bins correspond to residues in the RBD. The horizontal dotted line denotes the average number of amino-acid differences per bin across the protein-coding region of the whole viral genome. **d** Alignment of the sequences selected for each of the six classes. The "I" symbol demarcates every 10 residues in the alignment. **e** Representative images showing HCoV-229E infection of L-132 cells in the presence of: PBS, monoclonal antibody 9.8.E12 at two different concentrations, and monoclonal antibody 2.8H5 at two different concentrations (anti-HCoV-OC43 antibody). The nucleus is stained blue and green staining indicates viral infection. Magnification (×200) and scale bar = 10 μm. **f** Statistical quantification of the monoclonal antibody inhibition experiment. Error bars correspond to standard deviations obtained from three independent experiments

**Table 3 Surface plasmon resonance-binding data for the interaction between the six HCoV-229E RBDs and hAPN**

| Class | $k_{on}$ ($\times 10^5$ M$^{-1}$ s$^{-1}$) | $k_{off}$ (s$^{-1}$) | $K_d$ (nM) |
|---|---|---|---|
| I | 3.6 ± 0.5 | 0.16 ± 0.02 | 434 ± 63 |
| II | 3.3 ± 0.5 | 0.08 ± 0.02 | 246 ± 19 |
| III | 7.3 ± 1.4 | 0.08 ± 0.02 | 113 ± 2.3 |
| IV | 3.6 ± 0.5 | 0.10 ± 0.02 | 261 ± 24 |
| V | 4.8 ± 1.1 | 0.01 ± 0.01 | 27.0 ± 1.7 |
| VI | 8.5 ± 0.6 | 0.03 ± 0.01 | 37.4 ± 3.5 |

Values after ± correspond to the residual standard deviation reported by Scrubber 2. Two experiments were performed

remaining 55% (i.e., 279 Å$^2$) of the viral surface area buried is made up of 10 residues that differ in their variability and the role they play in complex formation (Supplementary Table 2).

**Loop variation leads to phylogenetic classes**. Phylogenetic analysis of the HCoV-229E RBD sequences found in the database showed that they segregate into six classes (Supplementary Fig. 6). Class I contains the ATCC-740 reference strain (originally isolated in 1967 and deposited in 1973) and related lab strains, while Classes II–VI, represent clinical isolates that have successively replaced each other in the human population over time since the 1970s. To characterize these classes, a representative sequence from each was selected; for Class I, the RBD of the reference strain, also used in our structural analysis, was selected. To simplify characterization, the RBDs of the other five classes were synthesized with the Class I sequence in all but the loop regions (Fig. 4d). As observed for Class I, the other RBDs do not bind to the hAPN mutant that introduces an N-glycan at Glu$^{291}$ (Supplementary Fig. 4D), an observation suggesting that they all bind at the same site on hAPN. The RBDs bound hAPN with an ~16-fold range in affinity ($K_d$ from ~30 to ~440 nM). These differences in affinity are largely a result of differences in $k_{off}$ with little difference in $k_{on}$ (Table 3 and Supplementary Fig. 7). Notably, the Class I RBD binds with the lowest affinity, while the RBDs from viral classes that have emerged most recently (Class V: viruses isolated in 2001–2004 and Class VI: viruses isolated in 2007–2015) bind with the highest affinity. For each of the six classes, Supplementary Table 2 shows the identity of the loop residues that have shown variation. Of those buried in the RBD–hAPN interface, residues 314, 404, and 407 are particularly noteworthy as they undergo considerable variation in amino-acid character. Residue 314, for example, accounts for 9% of the total buried surface area on complex formation and changes from Gly to Val to Pro in the transition from Classes I to VI. Variation of this sort provides insight into how changes in receptor-binding affinity might be mediated during the process of viral adaptation.

Each of the six RBD classes were also characterized using a neutralizing mouse monoclonal antibody (9.8E12) that we generated against the HCoV-229E reference strain (Class I). As shown in Fig. 4e, f, 9.8E12 inhibits HCoV-229E infection of the L132 cell-line. This antibody binds to the Class I RBD with a $K_d$ of 66 nM ($k_{on} = 6.3 \times 10^5$ M$^{-1}$ s$^{-1}$, $k_{off} = 0.041$ s$^{-1}$) and as shown by a competition binding experiment, it blocks the RBD–hAPN interaction (Supplementary Fig. 8A, B). In contrast, 9.8E12 shows no binding to the other five RBD classes at a concentration of 1 μM (Supplementary Fig. 8C), strong evidence that the receptor-binding loops of the Class I RBD are important for antibody binding and that loop variation can abrogate antibody binding. Consistent with this observation, non-conserved amino-acid changes both within and outside of the RBD–hAPN interface are observed across all classes (Supplementary Table 2).

## Discussion

Correlating structure and function with natural sequence data is a powerful means of studying viral adaptation and evolution. To this end, we have delimited the HCoV-229E RBD and determined its X-ray structure in complex with the ectodomain of its receptor, hAPN. We found that three extended loops on the RBD are solely responsible for receptor binding, and that these loops are highly variable among viruses isolated over the past 50 years. A phylogenetic analysis also showed that the RBDs of these viruses define six RBD classes whose viruses have successively replaced each other in the human population. The six RBDs differ in their receptor-binding affinity and their ability to be bound by a neutralizing antibody (9.8E12) and taken together, our findings suggest that the HCoV-229E sequence variation observed arose through adaptation and selection.

Antibodies that block receptor binding are a common route to viral neutralization and exposed loops are known to be particularly immunogenic[44]. Loop-binding neutralizing antibodies are elicited by the alphacoronavirus TGEV[40], and the receptor-binding loops of HCoV-229E mediate the binding of the neutralizing antibody, 9.8E12. As shown by the sequences of the viral isolates analyzed, the RBDs differ almost exclusively in their receptor-binding loops. 9.8E12 blocks the hAPN–RBD interaction and it can only bind to the RBD (Class I) found in the virus that elicited it. This observation shows that loop variability can abrogate neutralizing antibody binding. Indeed, the successive replacement or ladder-like phylogeny observed, when the sequence of the HCoV-229E RBD is analyzed, is characteristic of immune escape as shown by the influenza virus[45, 46]. Taken together, our results suggest that immune evasion contributes to if not explains the extensive receptor-binding loop variation shown by HCoV-229E over the past 50 years. HCoV-229E infection in humans does not provide protection against different isolates[37], and viruses that contain a new RBD class that cannot be bound by the existing repertoire of loop-binding neutralizing antibodies provide an explanation for this observation. Neutralizing antibodies that block receptor binding can also be thwarted by an increase in the affinity/avidity between the virus and its host receptor. Increased receptor-binding affinity/avidity allows the virus to more effectively compete with receptor blocking neutralizing antibodies, a mechanism thought to be important for evading a polyclonal antibody response[47]. In addition, an optimal receptor binding affinity is thought to exist in a given environment. As such, adaptation in a new species, changes in tissue tropism, and differences in receptor expression levels can all lead to changes in receptor binding affinity[29, 31, 48]. Taken together, the observation that the most recent RBD classes (Class V: viruses isolated in 2001–2004 and Class VI: viruses isolated in 2007–2015) show a ~16-fold increase in affinity for hAPN over that of Class I (viruses isolated in 1967) merits further study.

Recent cryoEM analysis has shown that the receptor-binding sites of HCoV-NL63, SARS-CoV, MERS-CoV, and by inference HCoV-229E, are inaccessible in some conformations of the pre-fusion S-protein trimer[21–25]. Although the ramifications of this structural arrangement are not yet clear, restricting access to the binding site has been proposed to provide a means of limiting B-cell receptor interactions against the receptor-binding site[23]. How this might work in mechanistic terms is also not clear given the need to bind receptor. However, in a simple model, the inaccessible S-protein conformation(s) would be in equilibrium with a less stable (higher energy) but accessible S-protein conformation(s). The energy difference between these conformations is a barrier to binding that decreases equally the intrinsic free energy of binding of both the viral receptor and the B-cell receptor and relative binding energies may be the key. Both soluble hAPN and

antibody 9.8E12 can inhibit HCoV–229E infection in a cell-based assay, an indication that their binding energies ($K_d$ of 430 and 66 nM, respectively) are sufficient to efficiently overcome the barrier to binding. However, B-cell receptors bind their antigens relatively weakly prior to affinity maturation[49] and they would be much less able to do so. The dynamics of the interconversion between accessible and inaccessible conformations may also be a factor in the recognition of inaccessible antibody epitopes[50, 51], and further work will be required to establish if and how restricting access to the receptor binding site enhances coronavirus fitness. The cryoEM structures also show that the receptor-binding loops make intra- and inter-subunit contacts in the inaccessible prefusion trimer. This suggests the intriguing possibility that the magnitude of the energy barrier, or the dynamics of the interconversion between accessible and inaccessible conformations, might be modulated by loop variation during viral adaption.

Immune evasion and cross-species transmission involve viral adaptation and we posit that the use of extended loops for receptor binding represents a strategy employed by HCoV–229E and the alphacoronaviruses to mediate the process. Such loops can tolerate insertions, deletions, and amino acid substitutions relatively free of the energetic penalties associated with the mutation of other protein structural elements. Indeed, our analysis of the six RBD classes shows that the receptor-binding loops possess a remarkable ability to both accommodate and accumulate mutational change while maintaining receptor binding. Among the six classes, 73% of the loop residues show change and only 45% of the receptor interface buried on receptor binding has been conserved. As we have shown, variation in the receptor-binding loops can abrogate neutralizing antibody binding and it will also increase the likelihood of acquiring new receptor interactions by chance. In this way, the selection of viral variants capable of immune evasion and/or cross-species transmission will be facilitated[27, 28, 52–54].

Cross-species transmission involves the acquisition of either a conserved (i.e., a similar interaction with a homologous receptor) or a non-conserved receptor interaction (i.e., an interaction with a non-homologous receptor, or an interaction at a new site on a homologous receptor) in the new host. HCoV–229E binds to a site on hAPN that differs from the site where PRCoV[40] binds to pAPN (Fig. 3a, b), and HCoV-NL63 is known to bind the non-homologous receptor, ACE2[55]. Clearly, conserved receptor interactions have not accompanied the evolution of these alphacoronaviruses (Fig. 3d–g). In mechanistic terms, receptor-binding loop variability and plasticity would facilitate the acquisition of both conserved and non-conserved receptor interactions. However, compared to conserved receptor interactions, the successful acquisition of non-conserved interactions would be expected to be relatively infrequent and more likely to require viral replication and mutation in the new host to optimize receptor-binding affinity.

Many coronaviruses have originated in bats[3, 4] and it is tempting to speculate that viral transmission between bats has facilitated the emergence of non-conserved receptor interactions. Bats account for ~20% of all mammalian species and they possess a unique ecology/biology that facilitates viral spread between them[56, 57]. Moreover, the barriers to viral replication in a new host are lower among closely related species[58, 59]. It follows that the viral replication required to optimize non-conserved receptor interactions in the new host would be facilitated by transmission between closely related bat species. By a similar reasoning, the use of conserved receptor interactions requiring little optimization would facilitate large species jumps. Several bat coronaviruses showing a high degree of sequence similarity with HCoV-229E

have recently been identified[60, 61] and an analysis of how they interact with bat APN will inform this discussion.

Predicting the emergence of new viral threats is an important aspect of public health planning[62] and our work suggests that RNA viruses that use loops to bind their receptors should be viewed as a particular risk. RNA viruses are best described as populations[34], and extended loops—inherently capable of accommodating and accumulating mutational change—will enable populations with loop diversity. Such populations will provide routes to escaping receptor loop-binding neutralizing antibodies, optimizing receptor-binding affinity, and acquiring new receptor interactions, interrelated processes that drive viral evolution and the emergence of new viral threats.

## Methods

**Protein expression and purification.** The soluble ectodomain of hAPN (residues 66–967) was expressed and purified from stably transfected HEK293S GnT1- cells (ATCC CRL-3022) as described previously[39]. The various soluble forms of the HCoV-229E S-protein were expressed and purified from stably transfected HEK293S GnT1-cells for X-ray crystallography, and from HEK293T (ATCC CRL-3216) and/or HEK293F (Invitrogen 51-0029) cells for cell-based and biochemical characterization, as described previously[63]. Point mutations were generated using the InFusion HD Site-Directed Mutagenesis protocol (Clontech). In all cases, the target proteins were secreted as N-terminal protein-A fusion proteins with a Tobacco Etch Virus (TEV) protease cleavage site following the protein-A tag. Harvested media was concentrated 10-fold and purified by IgG affinity chromatography (IgG Sepharose, GE). The bound proteins were liberated by on-column TEV protease cleavage and further purified by anion exchange chromatography (HiTrap Q HP, GE).

**Protein crystallization.** The RBD of the S-protein of HCoV-229E (residues 293–435) and the soluble ectodomain of hAPN (residues 66–967) were mixed in a ratio of 1.2:1 (RBD:hAPN) and the complex was purified by Superdex 200 (GE) gel filtration chromatography in 10 mM HEPES, 50 mM NaCl, pH 7.4. The complex was concentrated in gel filtration buffer to 10 mg/ml for crystallization trials. Crystals were obtained by the hanging drop method using a 1:1 mixture of stock protein and well solution containing 8% PEG 8000, 1 mM GSSG, 1 mM GSH, 5% glycerol, 1 μg/ml endo-β-$N$-acetylglucosaminidase A[64] and 100 mM MES, pH 6.5 at 298 K. Crystals were typically harvested after 3 days and flash-frozen with well solution supplemented with 22.5% glycerol as cryoprotectant.

**Data collection and structure determination.** Diffraction data were collected at the Canadian Light Source, Saskatoon, Saskatchewan (Beamline CMCF-08ID-1) at a wavelength of 0.9795 Å. Data were merged, processed, and scaled using HKL2000[65]; 5% of the data set was used for the calculation of $R_{free}$. Phases were obtained by molecular replacement using the human APN structure as a search model (PDB ID: 4FYQ) using Phaser in Phenix[66]. Manual building of the HCoV-229E RBD was performed using COOT[67]. Alternate rounds of manual rebuilding and automated refinement using Phenix were performed. Secondary structural restraints and torsion-angle non-crystallographic symmetry restraints between the three monomers in the asymmetric unit were employed. Ramachandran analysis showed that 96% of the residues are in the most favored region, with 4% in the additionally allowed region. Data collection and refinement statistics are found in Table 2. A stereo image of a portion of the electron density map in the HCoV–229E–hAPN interface is showed in Supplementary Fig. 9. Figures were generated using the program chimera[68]. Buried surface calculations were performed using the PISA server.

**Surface plasmon resonance binding assays.** Surface plasmon resonance (Biacore) assays were performed on CM-5 dextran chips (GE) covalently coupled to the ligand via amine coupling. The running and injection buffers were matched and consisted of 150 mM NaCl, 0.01% Tween-20, 0.1 mg/ml BSA, and 10 mM HEPES at pH 7.5. Response unit (RU) values were measured as a function of analyte concentration at 298 K. Kinetic analysis was performed using the global fitting feature of Scrubber 2 (BioLogic Software) assuming a 1:1 binding model. For experiments using hAPN as a ligand, between 300 and 400 RU were coupled to the CM-5 dextran chips. For experiments using 9.8E12, 1900 RU was immobilized.

**Viral inhibition assay.** HCoV-229E was originally obtained from the American Type Culture Collection (ATCC VR-740) and was produced in the human L132 cell line (ATCC CCL5) which was grown in minimum essential medium alpha (MEM-α) supplemented with 10% (v/v) FBS (PAA).

The L132 ($1 \times 10^5$) cells were seeded on coverslips and grown overnight in MEM-α supplemented with 10% (v/v) FBS. For inhibition assays in the presence of soluble hAPN, wild-type HCoV-229E ($10^{5.5}$ TCID$_{50}$) was pre-incubated with the fragment (residues 66–967) diluted in PBS for one hour at 37 °C before being

added to cells for 2 h at 33 °C. For inhibition assays in the presence of the soluble S-protein fragments, the different fragments, diluted in PBS, were added to cells and kept at 4 °C on ice for 1 h. Medium was then removed and cells were inoculated with wild-type HCoV-229E ($10^5$ $TCID_{50}$) for 2 h at 33 °C. For both inhibition assays, after the 2-h incubation period, medium was replaced and cells were incubated at 33 °C with fresh MEM-α supplemented with 1% (v/v) FBS for 24 h before being analyzed by an immunofluorescence assay (IFA).

Cells on the coverslips were directly fixed with 4% paraformaldehyde (PFA 4%) in PBS for 30 min at room temperature and then transferred to PBS. Cells were permeabilized in cold methanol (−20 °C) for 5 min and then washed with PBS for viral antigen detection. The S-protein-specific monoclonal antibody, 5-11H.6, raised against HCoV-229E (IgG1, produced in our laboratory by standard hybridoma technology), was used in conjunction with an AlexaFluor-488-labeled mouse-specific goat antibody (Life Technologies A-21202), for viral antigen detection[69]. After three washes with PBS, cells were incubated for 5 min with DAPI (Sigma-Aldrich) at 1 μg/ml to stain the nuclear DNA. To determine the percentage of L-132 cells positive for the viral S-protein, 15 fields containing a total of 150–200 cells were counted, at a magnification of ×200 using a Nikon Eclipse E800 microscope, for each condition tested in three independent experiments. Green fluorescent cells were counted as S-protein positive and expressed as a percentage of the total number of cells. Statistical significance was estimated by the analysis of variance (ANOVA) test and Tukey's test post hoc.

Monoclonal antibodies (IgG1, produced in our laboratory by standard hybridoma technology) raised against HCoV-229E (9.8E12) or HCoV-OC43 (2.8H5, negative control) that were found to be S-protein specific were tested in an infectivity neutralization assay. Wild-type HCoV-229E ($10^{5.5}$ $TCID_{50}$) was pre-incubated with the antibodies (1/100 of hybridoma supernatant) for 1 h at 37 °C before being added to L-132 cells for 2 h at 33 °C. Cells were washed with PBS and incubated at 33 °C with fresh MEM-α supplemented with 1% FBS (v/v) for 18 h before being analyzed by an immunofluorescence assay (IFA). Statistical significance was estimated by an ANOVA test, followed by post hoc Dunnett (two-sided) analysis.

**Comparative sequence analysis of HCoV-229E viral isolates**. The protein sequence of the HCoV-229E P100E isolate RBD (residues 293–435) was used to perform a search of the non-redundant protein sequence database using Blastp. Sequences were curated as of December 1, 2016. A total of 52 sequences were obtained with the GenBank Identifier numbers: NP_073551.1, AAK32188.1, AAK32189.1, AAK32190.1, AAK32191.1, CAA71056.1, CAA71146.1, CAA71147.1, ADK37701.1, ADK37702.1, ADK37704.1, BAL45637.1, BAL45638.1, BAL45639.1, BAL45640.1, BAL45641.1, AAQ89995.1, AAQ89999.1, AAQ90002.1, AAQ90004.1, AAQ90005.1, AAQ90006.1, AAQ90008.1, AFI49431.1, AFR45554.1, AFR79250.1, AFR79257.1, AGT21338.1, AGT21345.1, AGT21353.1, AGT21367.1, AGW80932.1, AIG96686.1 ABB90506.1, ABB90507.1, ABB90508.1, ABB90509.1, ABB90510.1, ABB90513.1. ABB90514.1, ABB90515.1, ABB90516.1, ABB90519.1, ABB90520.1, ABB90522.1, ABB90523.1, ABB90526.1, ABB90527.1, ABB90528.1, ABB90529.1, ABB90530.1, AOG74783.1. The 52 sequences were then aligned using Muscle[70] and the residue-specific sequence conservation index was mapped onto the surface of the RBD using the "render by conservation" tool in Chimera[68]. Percentage identity is mapped using a color scale with blue indicating 100% identity and red indicating 30% identity. The protein-coding regions of the eight sequences for which the entire genome were reported (GenBank Identifier numbers: NC_002645.1, JX503060.1, JX503061.1, KF514433.1, KF514430.1, KF514432.1, AF304460.1, and KU291448.1) were aligned using Muscle. The entire protein-coding region of the viral genome was treated as a continuous amino-acid string (8850 residues in total). Protein residues that were not identical among the eight sequences were counted as a difference and plotted in 100 residue bins. The sequence AAK32191.1 was chosen as the representative of Class I and the loop sequences of ABB90507.1, ABB90514.1, ABB90519.1, ABB90523.1, and AFR45554.1 were combined with the non-loop sequences of AAK32191.1 to generate the RBDs of Classes (II–VI), respectively.

**Data availability**. Coordinates and structure factors for the HCoV-229E RBD in complex with human APN were deposited in the protein data bank with PDB ID: 6ATK. The authors declare that all other data supporting the findings of this study are available within the article and its Supplementary Information files, or are available from the authors upon request.

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

## Acknowledgements
The work was supported by CIHR operating grants to J.M.R. and P.J.T. and a Canada Research Chair to P.J.T. The Canadian Light Source is acknowledged for synchrotron data collection.

## Author contributions
A.H.M.W. and J.M.R. designed the research. A.H.M.W., A.C.A.T., M.D., K.C. and C.S. performed the experiments. D.Z. and M.S. provided technical assistance. A.H.M.W. and J.M.R. wrote the manuscript with input from M.D. and P.J.T.

## Additional information

**Competing interests:** The authors declare no competing financial interests.

