## [Peer Review File · Nature Communications]

Reviewers' comments:

Reviewer #1 (Remarks to the Author):

This submission provides a structure for the receptor-binding domain (RBD) of the coronavirus 229E spike (S), in complex with its cellular receptor, human APN. The structure shows loops on the S RBD that are contacting the hAPN. Substitutions of several contact residues to alanine are shown to reduce RBD binding to the hAPN.

In 2012, other groups provided a structure for the RBD of a porcine coronavirus in complex with porcine (p)APN. The 229E RBD binds to hAPN at a site that is distinct and separate from the site on pAPN that binds porcine RBD. This is a new observation of interest to the field. The authors make reasonable speculations pertaining to this observation, for example, they appropriately highlight APN structures and APN conformations that can support infection by the human or porcine viruses.

There are 229E virus samples banked back to 1967. Sequencing showed that the S RBD loops hyper-vary throughout the natural history of the virus. The investigators showed that evolution of the loops, since 1967, conferred resistance to the binding of a neutralizing mAb that was raised against the 1967 isolate. They also demonstrated that the evolution of the loops correlated with increases in the affinity of the RBDs for hAPN. The latter observation of ever increasing hAPN binding during the last 50 years of 229E evolution is novel and is sure to be of interest to the field.

The study is relatively thorough, the results are statistically significant, and the text is concise. While it would obviously be more informative to have structures for the more recent 21st century S RBDs, and to also have neutralization data for the 21st century viruses, the report is quite complete as it stands. The report is liable to be read and cited by several groups that are currently using cryo-EM to obtain complete CoV S protein structures. Cryo-EM images of 229E S trimers will likely be obtained very soon, and this thorough analysis of a 229E S RBD in complex with hAPN will be valuable in generating a more complete view of virus entry.

There are a couple of comments.

1. The discussion emphasizes antibodies as selective agents driving evolution of RBD loops. This is reasonable but it does not seem to account for the evolution of RBDs with increased hAPN affinity. The possible connections between antibody escape and receptor affinity, touched upon on lines 238-240, might deserve additional exposition. At present, the discussion of this interesting issue appears incomplete, too brief for many readers.
2. The discussion on lines 248-252 are somewhat unclear (at least to this reviewer). The cryo-EM structures of S ectodomain trimers do indicate that S RBDs are buried within the trimer interfaces, but it does not yet seem clear what "drives" the S protein conformational changes that then present the RBDs to cell receptors. Perhaps RBDs spontaneously are in the "standing" receptor-accessible position, and then antibodies or receptors can bind. It is

not clear what is meant by soluble APN and antibody affinities being “sufficient to overcome the barrier to binding”. The investigators seem to have an important and potentially insightful discussion point here that, in this reviewer’s opinion, deserves further clarification.

Reviewer #2 (Remarks to the Author):

Wong et al. present a new structure of a 229E strain Corona Virus bound to its aminopeptidase N receptor. The structure is of modest resolution, but appears to be well determined. The validation report shows excellent agreement between structure and data and agreement with chemical knowledge. The structure is, however, determined to modest (3.5Å) resolution. This doesn't directly compromise any claims made by the authors. I'm unable to enthusiastically support publication in Nature Communications because the main results from the study appear, to me, to fall short of the manuscripts I've read in this journal. The authors make a strong argument, based on very detailed mutations and affinity measurements that the entirety of the binding energy comes from the 3 loops they characterize. Further, they show that Corona Viruses engage the same receptor differently across strains. This is perhaps the most surprising result. Influenza, for example, doesn't do this, although HIV does. The phylogeny presented doesn't appear to be enhanced by the structural insights. It isn't particularly surprising that a neutralizing antibody is strain specific. The discussion is interesting, although I'd caution the authors to avoid a thermodynamic argument concerning how conformational rearrangements can limit receptor access. This may be an important aspect of immune evasion for Corona viruses, but some understanding of the kinetics involved is needed. In short, I think this manuscript will be received very well at a microbiology journal.

Response to Referees' comments:

Reviewer #1 (Remarks to the Author):

This submission provides a structure for the receptor-binding domain (RBD) of the coronavirus 229E spike (S), in complex with its cellular receptor, human APN. The structure shows loops on the S RBD that are contacting the hAPN. Substitutions of several contact residues to alanine are shown to reduce RBD binding to the hAPN.

In 2012, other groups provided a structure for the RBD of a porcine coronavirus in complex with porcine (p)APN. The 229E RBD binds to hAPN at a site that is distinct and separate from the site on pAPN that binds porcine RBD. This is a new observation of interest to the field. The authors make reasonable speculations pertaining to this observation, for example, they appropriately highlight APN structures and APN conformations that can support infection by the human or porcine viruses.

There are 229E virus samples banked back to 1967. Sequencing showed that the S RBD loops hyper-vary throughout the natural history of the virus. The investigators showed that evolution of the loops, since 1967, conferred resistance to the binding of a neutralizing mAb that was raised against the 1967 isolate. They also demonstrated that the evolution of the loops correlated with increases in the affinity of the RBDs for hAPN. The latter observation of ever increasing hAPN binding during the last 50 years of 229E evolution is novel and is sure to be of interest to the field.

The study is relatively thorough, the results are statistically significant, and the text is concise. While it would obviously be more informative to have structures for the more recent 21st century S RBDs, and to also have neutralization data for the 21st century viruses, the report is quite complete as it stands. The report is liable to be read and cited by several groups that are currently using cryo-EM to obtain complete CoV S protein structures. Cryo-EM images of 229E S trimers will likely be obtained very soon, and this thorough analysis of a 229E S RBD in complex with hAPN will be valuable in generating a more complete view of virus entry.

There are a couple of comments.

1. The discussion emphasizes antibodies as selective agents driving evolution of RBD loops. This is reasonable but it does not seem to account for the evolution of RBDs with increased hAPN affinity. The possible connections between antibody escape and receptor affinity, touched upon on lines 238-240, might deserve additional exposition. At present, the discussion of this interesting issue appears incomplete, too brief for many readers.

[R1-1] Thanks very much for your comments regarding the importance of our work and for this suggestion. We have now expanded the relevant section in the revised manuscript. Increased receptor binding affinity allows the virus to more effectively compete with receptor-blocking neutralizing antibodies. As such, increased affinity/avidity is thought to provide a route to thwarting a polyclonal antibody response that would be otherwise difficult to overcome. We also now expand on the statement that an optimal receptor binding affinity/avidity is thought to exist. Specifically, we

now mention that cross-species transmission, changes in tissue tropism and differences in receptor expression levels can all lead to changes in receptor binding affinity.

The revised section appears as follows with changes highlighted:

Lines (242-256): Taken together, our results suggest that immune evasion **contributes to if not** explains the extensive receptor-binding loop variation shown by HCoV-229E over the past fifty years. HCoV-229E infection in humans does not provide protection against different isolates [38], and viruses that contain a new RBD class that cannot be bound by the existing repertoire of loop-binding neutralizing antibodies provide an explanation for this observation. **Neutralizing antibodies that block receptor binding can also be thwarted by an increase in the affinity/avidity between the virus and its host receptor. Increased receptor binding affinity/avidity allows the virus to more effectively compete with receptor blocking neutralizing antibodies, a mechanism thought to be important for evading a polyclonal antibody response [48]. In addition, an optimal receptor binding affinity is thought to exist in a given environment. As such, adaptation in a new species, changes in tissue tropism, and differences in receptor expression levels can all lead to changes in receptor binding affinity [30, 32, 49].** Taken together, the observation that the most recent RBD classes (Class V: viruses isolated in 2001 - 2004 and Class VI: viruses isolated in 2007 - 2013) show a ~16-fold increase in affinity for hAPN over that of Class I (viruses isolated in 1967) merits further study.

2. The discussion on lines 248-252 are somewhat unclear (at least to this reviewer). The cryo-EM structures of S ectodomain trimers do indicate that S RBDs are buried within the trimer interfaces, but it does not yet seem clear what “drives” the S protein conformational changes that then present the RBDs to cell receptors. Perhaps RBDs spontaneously are in the “standing” receptor-accessible position, and then antibodies or receptors can bind. It is not clear what is meant by soluble APN and antibody affinities being “sufficient to overcome the barrier to binding”. The investigators seem to have an important and potentially insightful discussion point here that, in this reviewer’s opinion, deserves further clarification.

[R1-2] Thanks for drawing attention to a possible source of confusion. With the use of a model we now explain how inaccessible conformations - in equilibrium with accessible conformations - can reduce the extent of binding to B-cell receptors while still allowing adequate binding for high affinity receptors and antibodies. As noted by referee 2, kinetic considerations may also be important. We agree and for completeness have now mentioned this and cited two papers that deal with the current thinking on how viral glycoprotein dynamics might impact on antibody interactions.

The revised section appears as follows with changes highlighted:

Lines (257-278): Recent cryoEM analysis has shown that the receptor binding sites of HCoV-NL63, SARS-CoV, MERS-CoV, and by inference HCoV-229E, are inaccessible in some conformations of the pre-fusion S-protein trimer [22-26]. Although the ramifications of this structural arrangement are not yet clear, restricting access to the binding site has been proposed to provide a means of limiting B-cell receptor interactions against the receptor binding site [24]. **How this might work in**

mechanistic terms is also not clear given the need to bind receptor. However, in a simple model, the inaccessible S-protein conformation(s) would be in equilibrium with a less stable (higher energy) but accessible S-protein conformation(s). The energy difference between these conformations is a barrier to binding that decreases equally the intrinsic free energy of binding of both the viral receptor and the B-cell receptor and relative binding energies may be the key. Both soluble hAPN and antibody 9.8E12 can inhibit HCoV-229E infection in a cell-based assay, an indication that their binding energies (K_{DS} of 430 nM and 66 nM, respectively) are sufficient to efficiently overcome the barrier to binding. However, B-cell receptors bind their antigens relatively weakly prior to affinity maturation [50] and they would be much less able to do so. The dynamics of the interconversion between accessible and inaccessible conformations may also be a factor in the recognition of inaccessible antibody epitopes [51, 52], and further work will be required to establish if and how restricting access to the receptor binding site enhances coronavirus fitness. The cryoEM structures also show that the receptor-binding loops make intra- and inter-subunit contacts in the inaccessible prefusion trimer. This suggests the intriguing possibility that the magnitude of the energy barrier, or the dynamics of the interconversion between accessible and inaccessible conformations, might be modulated by loop variation during viral adaptation.

Reviewer #2 (Remarks to the Author):

Wong et al. present a new structure of a 229E strain Corona Virus bound to its aminopeptidase N receptor. The structure is of modest resolution, but appears to be well determined. The validation report shows excellent agreement between structure and data and agreement with chemical knowledge. The structure is, however, determined to modest (3.5Å) resolution. This doesn't directly compromise any claims made by the authors. I'm unable to enthusiastically support publication in Nature Communications because the main results from the study appear, to me, to fall short of the manuscripts I've read in this journal. The authors make a strong argument, based on very detailed mutations and affinity measurements that the entirety of the binding energy comes from the 3 loops they characterize. Further, they show that Corona Viruses engage the same receptor differently across strains. This is perhaps the most surprising result. Influenza, for example, doesn't do this, although HIV does. the phylogeny presented doesn't appear to be enhanced by the structural insights. It isn't particularly surprising that a neutralizing antibody is strain specific. The discussion is interesting, although I'd caution the authors to avoid a thermodynamic argument concerning how conformational rearrangements can limit receptor access. This may be an important aspect of immune evasion for Corona viruses, but some understanding of the kinetics involved is needed. In short, I think this manuscript will be received very well at a microbiology journal.

[R2-1] Thanks for comments on the quality and thoroughness of our work. With regard to the comment on phylogeny and structure, receptor and neutralizing antibody complexes (Fabs) with the various RBD classes will ultimately be required to understand the details. However, the importance of our work stems from the finding that three extended loops are solely responsible for receptor binding and that these loops have undergone extensive natural sequence variation over the past 50 years. As such, it provides a striking demonstration of an interplay between fundamental protein

structural principles - the inherent ability of extended loops to accumulate and accommodate change - and error-prone RNA replication, the basis for viral adaptation and selection. We may not know why a particular strain can't be bound by a given neutralizing antibody, but we have provided important insight into the structural basis for the adaptation process involved. Moreover, we show that both the abrogation of antibody binding and increases in receptor binding affinity may be involved. Indeed, we feel that the strength of our interdisciplinary approaches, analysis and insights, provide the basis for publication in Nature Communications.

Nevertheless, we now see that we could have provided some additional analysis of the loop sequences of the various phylogenetic classes in light of our structural results. Specifically, Table S3 has been expanded to list all loop residues that show change across the 6 RBD classes; as in the previous version of our manuscript the table also indicates which residues are buried by hAPN in the class I RBD-hAPN complex. The reader can now examine the temporal change (ie. from Class I to Class VI) at each loop position both within and outside of the receptor binding interface. With reference to the table, we highlight in the revised manuscript non-conserved residue changes in the receptor binding interface that likely contribute to the receptor binding affinity differences shown by the RBD's of the six classes. We also refer to the table when pointing out that residue changes both within and outside of the interface have the potential to abrogate antibody binding.

The revised section appears as follows with changes highlighted:

Lines 204-210: For each of the 6 classes, Table S3 shows the identity of the loop residues that have shown variation. Of those buried in the RBD-hAPN interface, residues 314, 404, and 407 are particularly noteworthy as they undergo considerable variation in amino acid character. Residue 314, for example, accounts for 9% of the total buried surface area on complex formation and changes from Gly to Val to Pro in the transition from Class I to VI. Variation of this sort provides insight into how changes in receptor binding affinity might be mediated during the process of viral adaptation.

Lines 219-220: Consistent with this observation, non-conserved amino acid changes both within and outside of the RBD-hAPN interface are observed across all classes (Table S3).

[R2-2] Thanks for the endorsement of our discussion. With regard to kinetic considerations, we did not mean to suggest that they might not be important. As detailed above [R1-2], we have expanded and clarified our thermodynamic explanation of how inaccessible conformations can differentially limit weak B-cell receptor interactions relative to high affinity receptor or antibody interactions. In addition, we now point out that viral glycoprotein kinetics/dynamics may also be important in antibody recognition and neutralization and we cite two references to recent work in this area.

REVIEWERS' COMMENTS:

Reviewer #1 (Remarks to the Author):

The authors have responded to my comments by adding text in the Results and Discussion sections. Their additions address my concerns. The additions increase clarity and improve the manuscript.

Reviewer #2 (Remarks to the Author):

I appreciate the changes the authors have made to the supplemental figure 3 and the way it has been incorporated into the manuscript.

My limited enthusiasm did not stem from concerns about the quality or clarity of the manuscript. I'm fully supportive of the decision to accept this manuscript for publication.